# Immediate Effects of Myofascial Release Treatment on Lumbar Microcirculation: A Randomized, Placebo-Controlled Trial

**DOI:** 10.3390/jcm12041248

**Published:** 2023-02-04

**Authors:** Andreas Brandl, Christoph Egner, Rüdiger Reer, Tobias Schmidt, Robert Schleip

**Affiliations:** 1Department of Sports Medicine, Institute for Human Movement Science, Faculty for Psychology and Human Movement Science, University of Hamburg, 20148 Hamburg, Germany; 2Department for Medical Professions, Diploma Hochschule, 37242 Bad Sooden-Allendorf, Germany; 3Osteopathic Research Institute, Osteopathie Schule Deutschland, 22297 Hamburg, Germany; 4Institute of Interdisciplinary Exercise Science and Sports Medicine, MSH Medical School Hamburg, 20457 Hamburg, Germany; 5Conservative and Rehabilitative Orthopedics, Department of Sport and Health Sciences, Technical University of Munich, 80333 Munich, Germany

**Keywords:** microcirculation, thoracolumbar fascia, fascia morphology, physical activity, myofascial release

## Abstract

(1) Background: Inflammatory processes in the thoracolumbar fascia (TLF) lead to thickening, compaction, and fibrosis and are thought to contribute to the development of nonspecific low back pain (nLBP). The blood flow (BF) of fascial tissue may play a critical role in this process, as it may promote hypoxia-induced inflammation. The primary objective of the study was to examine the immediate effects of a set of myofascial release (MFR) techniques on the BF of lumbar myofascial tissue. The secondary objectives were to evaluate the influence of TLF morphology (TLFM), physical activity (PA), and body mass index (BMI) on these parameters and their correlations with each other. (2) Methods: This study was a single-blind, randomized, placebo-controlled trial. Thirty pain-free subjects (40.5 ± 14.1 years) were randomly assigned to two groups treated with MFR or a placebo intervention. Correlations between PA, BMI, and TLFM were calculated at baseline. The effects of MFR and TLFM on BF (measured with white light and laser Doppler spectroscopy) were determined. (3) Results: The MFR group had a significant increase in BF after treatment (31.6%) and at follow-up (48.7%) compared with the placebo group. BF was significantly different between disorganized and organized TLFM (*p* < 0.0001). There were strong correlations between PA (r = −0.648), PA (d = 0.681), BMI (r = −0.798), and TLFM. (4) Conclusions: Impaired blood flow could lead to hypoxia-induced inflammation, possibly resulting in pain and impaired proprioceptive function, thereby likely contributing to the development of nLBP. Fascial restrictions of blood vessels and free nerve endings, which are likely associated with TLFM, could be positively affected by the intervention in this study.

## 1. Introduction

The thoracolumbar fascia (TLF) is a salient central anatomic structure in the dorsal trunk region. Therefore, many authors consider this structure as a potential trigger for nonspecific low back pain (nLBP) [1,2,3,4,5,6,7,8]. This multilayered fascia consists of dense aponeuroses that attach to the posterior vertebral processes as well as to the supraspinal ligaments in the thoracolumbar spine area. These aponeuroses can withstand loads [1] and transmit forces [9]. In addition, they have thin layers of loose connective tissue that separate these aponeuroses as well as the underlying epimysial muscle layers and support shear mobility, which plays an important role in the movement mechanics of the trunk [10]. In addition, subcutaneous fascial bands have been found to mechanically connect skin, the subcutaneous layers, and deeper muscles [11]. From a biomechanical point of view, the range of motion (ROM) represents the maximum distance over which a joint can be moved and, in the trunk, depends to a large extent on the mechanical properties of the fascia, since it is tightly connected to the underlying muscle along its entire peripheral length [12]. In a morphological sense, fascia also connects muscles that are arranged in series [9]. Because of these relationships between fascia and muscle, a stiff TLF can directly limit ROM [10].

There are some hypotheses suggesting that inflammatory processes within the TLF lead to thickening, densification, and fibrosis [2,10,13,14]. Bishop et al. [15] investigated the influence of injury to the TLF and restriction in motion and found that the combination of these factors leads to significant fibrosis and thickening of the TLF. In addition, recent studies have shown that the TLF has extensive sensory innervation, including small-caliber nociceptors that can be activated by mechanical stimulation and are apparently affected by morphological changes [3,5,6,16]. The TLF is also thought to play an important role in lumbar proprioception [5,17]. There is increasing evidence that not only thickening but also ultrasonic echogenicity and disorganization of the fascia are critical factors in the development of nLBP, ROM limitations, and sensory disturbances [2,18,19,20,21]. Therefore, these morphological differences in TLF characteristics may reflect, first, how mechanical forces are transmitted across different tissues, second, are part of the explanation for limited trunk ROM, and third, cause sensory disturbances (e.g., pain from nociceptor stimulation or inadequate proprioceptive response). De Coninck et al. [20] developed a scoring scheme to critically evaluate the morphological characteristics of the TLF at four levels ranging from 1 = “very disorganized” to 4 = “very organized” and found that even medical practitioners inexperienced in the evaluation of ultrasound images achieved sufficient reliability in assessing the four groups of TLF organization. Although expert consensus on the specific classification criteria is still pending, this scheme represents the first morphological assessment protocol for TLF.

Blood circulation in fascial tissue may play a critical role in maintaining the biomechanical, proprioceptive, and nociceptive function of the TLF [5,22,23,24]. In particular, a decrease in blood flow with subsequent deoxygenation may be a trigger for fascial degeneration (e.g., stiffening, fibrosis, or thickening). In addition, hypoxia-related proteins are mediators of cell inflammation and apoptosis and cause a significant shift in collagen matrix and expression, which may also promote fascial degeneration [22]. On the other hand, fascial restrictions obviously decrease blood flow by vascular compression [24]. It is known from previous studies that fascial blood flow and oxygen saturation are dependent on individual physical activity (PA), age, and body mass index (BMI) [10,20,22,23,25,26]. Neurological factors also play an important role in the blood supply to fascia [5,6,16]. Mense [5] found a large amount of postganglionic sympathetic fibers, most of which probably act as vasoconstrictors by innervating the arterioles of the TLF. Free vascular nerve endings are able to release neuropeptides and neurotrophins after mechanical stimuli (e.g., myofascial release (MFR) treatment) and lead to vasodilation by increasing capillary permeability. This makes blood circulation an interesting variable to study in terms of the relationship between PA, BMI, TLF morphology (TLFM), and its interaction with manual treatment.

MFR techniques are widely used in manual medicine with the aim of restoring the optimal length of myofascial tissue structures, improving their function and reducing their pain [27,28]. They are also commonly used by physiotherapists in the treatment of nLBP. However, due to the poor quality of most current MFR studies, they also urged the need for future studies of high quality. In MFR treatment, the myofascial tissue is treated with a mechanical shearing motion (combination of compression and stretching) with low force and slow speed [27]. This is thought to result in a lasting change in the morphology of the fascia and also its hydration [25,26,29,30], because fascial tissue response to balanced, sustained stretching is more likely than to intermittent, uneven loads [31]. In addition, the numerous free nerve endings that act as proprioceptors, nociceptors, or nerve structures that innervate blood vessels could trigger a range of neuromuscular and neurovascular reflexes [5,32].

To the authors’ knowledge, there are no studies to date that have specifically investigated the influence of myofascial morphology or manual treatment on the blood microcirculation of TLF in vivo, which is the aim of the present work.

The first hypothesis (primary objective) was that the presumed effect of MFR on fascial tissue would lead to an increase in blood flow (BF), oxygen saturation of the hemoglobin (SO_2_), and relative hemoglobin concentration in tissue (rHb) after treatment with a series of MFR techniques. To determine whether a possible BF increase occurs only immediately after the intervention or whether the effect lasts longer, a 60 min follow-up was considered. Previous studies reported that temperature, and thus presumed skin BF, decreases after this period [33,34].

The authors further hypothesized (secondary objectives) that TLFM, besides PA and BMI, has an influence on blood circulation in the myofascial lumbar tissue. Therefore, one objective was to investigate the influence of TLFM, PA, and BMI on BF, SO_2_, and rHb. Another objective was to investigate the correlations between PA, BMI, and TLFM.

## 2. Materials and Methods

### 2.1. Study Design Overview

The study was a single-blind, randomized, placebo-controlled trial with two groups. Measurements were taken before and after the intervention and at a 60 min follow-up according to the SPIRIT guidelines [21]. The study protocol was prospectively registered with the German Clinical Trials Register (DRKS00028780) on 8 April 2022. The study has been reviewed and approved by the ethical committee of the Diploma Hochschule, Germany (Nr. 1014/2021), has been carried out in accordance with the declaration of Helsinki, and has obtained written informed consent from the participants [35].

### 2.2. Setting and Participants

The study was conducted in an osteopathic practice, in a medium-sized city in southern Germany. The number of participants per group was calculated based on the MFR effect size of a previous study (partial η^2^ = 0.22, α err = 0.05, 1-β err = 0.9) and set at 15 per group [4,36]. The acquisition was carried out via direct contact, a notice board, and the distribution of information material in the practice. The study design envisaged carrying out the study in a running practice, which is why the participants were continuously recruited during the study period. All test persons received a voucher for a preventive service of their own choice in the amount of 30 euros.

Inclusion criteria were: (a) participants without a history of lumbar back pain [37] or acute musculoskeletal disorders (symptom-free for at least four weeks); (b) a BMI between 18 and 29.9; (c) a maximum dermis (DER) and subcutaneous adipose tissue (SAT) thickness of 7 mm in the measurement area of the ultrasound (Figure 1); (d) female or male subjects aged 18 to 60 years; (e) prone position for 15 min must be pain-free for the subjects.

Exclusion criteria were: (a) generally valid contraindications to physiotherapeutic and osteopathic treatments of the lumbar spine and pelvis; (b) rheumatic diseases; (c) taking medication that affects blood coagulation; (d) taking muscle relaxants; (e) skin changes (e.g., neurodermatitis, psoriasis, urticaria, decubitus ulcers); (d) surgery or other scars in the lumbar region between Th12 and S1; (e) pregnancy.

### 2.3. Randomization and Interventions

The volunteers were first screened for eligibility by the investigator (AB). They completed the International Physical Activity Questionnaire (IPAQ) [38] and were examined by ultrasound for TLF morphology. Subjects were then covertly assigned to the MFR or placebo group (PLC) using block randomization. The Internet-based randomization was carried out with the application Research Randomizer, version 4.0 [25]. Subjects were not given any information by the investigator regarding their group membership or the intervention that was being delivered. After a 10 min rest period on a therapy table in the prone position, the investigator then performed the first measurement of blood flow data (BFD). The subjects received the respective group-specific intervention from an osteopath who had more than 15 years of professional experience in manual therapy and a master’s degree. This was followed by the post-intervention measurement and a 60 min follow-up examination. For this purpose, the subjects continued to lie in a relaxed prone position on the therapy table. The examination and treatment were carried out by the owner of the individual practice for osteopathy (AB).

#### 2.3.1. Myofascial Release Intervention

The MFR group received an MFR treatment protocol that included four previously described techniques [4,39,40,41], as follows: (1) Sustained manual pressure to the lateral raphe (Figure 2a), performed with the therapist’s fingertips 1–4. (2) Lateral stretching of the TLF (Figure 2b), performed with the therapist’s hands. (3) Longitudinal glide along the lumbar paravertebral muscles (Figure 2c), performed with the therapist’s open fist. (4) Longitudinal stretch of the TLF (Figure 2d), performed with the therapist’s hands. (5) Unilateral longitudinal stretch of the TLF (Figure 2e), performed with the therapist’s hands.

Here, except for technique 1, the subject is in a prone position with the arms on the sides of the body and the legs parallel to each other (Figure 2b–e). The head is in a neutral position, the face lies in a recess in the head section of the therapy table. The patient is undressed to such an extent that the TLF between Th12 and S1 is accessible. In technique 1, the patient lies in the lateral position with the shoulders flexed 90° and the hips flexed 30° (Figure 2a). The therapist stands contralateral to the side to be treated, at the level of the subject’s iliac crest. The therapist initiates at all techniques a direct stretch of the fascia to a noticeable tissue resistance. The therapist then follows the creep of the myofascial tissue to initiate further stretching of the TLF [1]. The applied force on the tissue is only moderate, ranging from 25 to 35 N. The usually applied force during an MFR treatment was previously evaluated by the therapist using a precision scale. The duration of each technique is 60 to 90 s. However, the decisive factor for the effect is not so much the time period over which the technique is practiced, but the occurrence of a myofascial release. Ajimsha et al. [27] define this as the restoration of the optimal length of myofascial tissue structures, their functional improvement, and the reduction of pain in them. All techniques were applied on both sides of the TLF. Therefore, the whole intervention lasted between 10 and 12 min.

#### 2.3.2. Control Group

The subjects in the PLC group were in the same position as those in the MFR group (Figure 2a–e). Instead of the MFR treatment protocol, sham treatment was applied only with minimal pressure between 4 and 6 N to the same areas treated in the MFR group. The duration was 90 s for each area on each TLF side, except for 5, which was touched unilaterally.

### 2.4. Outcomes

#### 2.4.1. Thoracolumbar Fascia Morphology

Ultrasound images (Philips Lumify with linear transducer L12-4, B-Mode, 12 MHz, 3.5 cm depth; Philips Ultrasound, Inc., Bothell, WA, USA) were acquired 2 cm lateral to the lumbar interspinous level 2–3 according to Langevin et al. [18] because the fascial planes are most parallel to the skin at this level. Therefore, the interspinous ligament between lumbar vertebrae 2 and 3 and the superficial border of the posterior paraspinal muscles were imaged according to the validated protocol of Stokes et al. [42]. The ultrasound image section was placed as close as possible to the thoracolumbar complex (Figure 1). Longitudinal images were then taken 2 cm lateral to the disc space between lumbar vertebrae 2 and 3. The side of the body to be measured was determined by a random number generator. The investigator (A.B.) then ensured that the inclusion criteria of a maximum DER and SAT thickness of 7 mm were met.

Subsequently, the TLFM was divided into 4 groups according to De Coninck et al. [20]: group 1: very disorganized, group 2: somewhat disorganized, group 3: somewhat organized, group 4: very organized. Three different raters with at least 10 years of experience in evaluating ultrasound images (two physicians and one osteopath with a master’s degree) independently assessed TLFM in a blinded fashion with respect to the study participants and to the other raters. A Likert scale was used with rating points from 1 to 10; point 1 was labeled “very disorganized” and point 10 was labeled “very organized”; intermediate points were numbered but not labeled. Raters were familiarized with the definition of TLF organization and disorganization. For example, “very organized” was defined as “being able to draw a rectangular box around the hyperechogenic area of the TLF”. The mean of the three scores was used for further analysis. The participants’ ratings on the scans were then divided into the four groups. Group 1 included all scans with a median score of 1 to 3. Group 2 included all median scores of 4 to 5. Group 3 included all median scores of 6 to 7. Group 4 included all median scores of 8 to 10 (for review, see De Coninck et al. [20]). 

De Coninck et al. [20] attest a degree of agreement, with a Krippendorff alpha of 0.61, and a degree of consistency, with a Conbach’s alpha of 0.98, with this procedure, even for inexperienced examiners. Almazán-Polo et al. [43] found excellent intra-rater reliability. The intraclass correlation coefficient (ICC) value for the right TLF was ICC = 0.968, and for the left TLF, it was ICC = 0.955.

#### 2.4.2. Blood Flow Measurements

To obtain a reproducible position for measuring blood flow in the lumbar myofascial tissue, the previously mentioned ultrasound measurement point at the intersection of the lateral edge of the transducer and the center of the transducer was marked with skin markers. This position also allows measurement of thoracolumbar fascia blood flow in addition to muscle tissue.

Measurement of BFD data was performed using the O2C device (LEA Medizintechnik, Giessen, Germany). Based on an optical fiber probe, the O2C is a noninvasive device for measuring blood flow and oxygenation of subcutaneous tissue to a depth of 8 mm. Multiple fiber probes include both white light (wavelengths of 500–800 nm) and laser Doppler (830 nm and 30 mW) spectroscopy. Using white light spectroscopy allows the determination of hemoglobin parameters, SO_2_ and rHb. Laser Doppler flowmetry provides the ability to determine perfusion parameters in tissues as it detects all moving erythrocytes. The number of moving erythrocytes is then processed in combination with blood flow velocity to provide the parameter BF, which is expressed in arbitrary units (AU). In numerous studies, this device has been shown to be reliable and valid [23,44].

For standardization, environmental factors such as light and temperature were kept constant. An experienced researcher (A.B.) performed all measurements. The O2C fiber probe was fixed pressure-free with a transparent double-sided adhesive film (Lea Medizintechnik, Giessen, Germany). The stability of the measurement was investigated by collecting BFD from ten subjects at a 10 min interval and calculating the intra-rater reliability. The ICC value for this was excellent (ICC = 0.99).

### 2.5. Statistics

For all parameters, the standard deviation (SD), the mean, and the 95% confidence interval (95% CI) were determined. There were no outliers in the data. The outcome variables were normally distributed as assessed by the Shapiro–Wilk test (*p* > 0.05). The homogeneity of the error variances between the groups was fulfilled for all these variables according to Levene’s test (*p* > 0.05). 

The ICC within the rater of the BFD and between raters of the TLFM and their 95% CI were calculated using the R package “irr” version 0.84.1 based on a 2-way mixed-effects model with absolute agreement. Resulting ICC values were interpreted according to Fleiss [45] as “poor” (<0.4), “fair to good” (0.4 to 0.75), and “excellent” (>0.75). 

To identify the difference between MFR and PLC groups in BF, SO_2_, and rHb, two-way ANOVAs were carried out. Post hoc analysis was conducted using Tukey’s HSD test. To ensure comparability between groups, differences in BF, SO_2_, and rHb between the MFR and PLC groups were tested at baseline with an independent Student’s t-test. Three-way ANOVAs were conducted to determine the effects of TLFM, IPAQ and BMI on BF, SO_2_, and rHb. Post hoc analysis was conducted using Tukey´s HSD test.

Somers’ D was performed to determine associations between TLFM and IPAQ, and Spearman’s rank correlation for associations between TLFM and BMI, as well as TLFM. Resulting values were interpreted according to Cohen [46] as “weak” (>0.09, <0.30), “medium” (>0.29, <0.50), and “strong” (≥0.50).

The significance level was set at *p* = 0.05.

Libreoffice Calc version 6.4.7.2 (Mozilla Public License v2.0) was used for the descriptive statistics. The inferential statistics were carried out with the software R, version 3.4.1 (R Foundation for Statistical Computing, Vienna, Austria). 

## 3. Results

The anthropometric data and baseline characteristics are shown in Table 1. Of 38 subjects screened between 11 April 2022 and 29 April 2022, 30 met the eligibility criteria (Figure 3). No subject was unblinded accidentally or in any other way.

### 3.1. Primary Objective: Difference between Myofascial Release Treatment and Placebo Intervention

According to Student’s t-test, there were no group differences between the MFR and PLC groups at baseline in BF, t(28) = −1.19, *p* = 0.243, SO_2_, t(28) = −0.53, *p* = 0.603, and rHb, t(28) = −1.31, *p* = 0.208.

A two-way ANOVA determined significant interaction between the study groups and measurement timepoints (MP) on BF, F(1, 56) = 5.24, *p* = 0.026, partial η^2^ = 0.09. Consequently, an analysis of simple main effects for the study groups was performed with statistical significance receiving a Bonferroni adjustment. There was a statistically significant difference in mean BF scores for t_1_ (F(1, 56) = 35.8, *p* < 0.0001) and t_2_ (F(1, 56) = 85.0, *p* < 0.0001) between study groups. According to Tukey’s HSD test, the MFR group showed significantly higher BF for t_1_ (M_Diff_ = 31.6, 95%-CI[41.5, 21.7], *p* < 0.0001) and t_2_ (M_Diff_ = 48.7, 95%-CI[60.4, 37.0], *p* < 0.0001) compared with the control group.

A two-way ANOVA determined no significant interaction between the study groups and MP on SO_2_, F(1, 56) = 0.47, *p* = 0.496, partial η^2^ = 0.01. There was a significant main effect for the study groups on SO_2_, F(1, 56) = 7.26, *p* = 0.009, partial η^2^ = 0.11, but not for MP, F(1, 56) = 0.79, *p* = 0.379, partial η^2^ < 0.01. According to Tukey’s HSD test, the MFR group showed significantly higher SO_2_ for t_2_ (M_Diff_ = 11.7, 95%-CI[23.1,0.4], *p* = 0.043), but not for t_1_ (M_Diff_ = 6.97, 95%-CI[15.6, –1.69], *p* = 0.11) compared with the control group.

A two-way ANOVA determined no significant interaction between the study groups and MP on rHb, F(1, 56) = 0.003, *p* = 0.959, partial η^2^ < 0.01. There was a significant main effect for the study groups on rHb, F(1, 56) = 20.06, *p* = 0.037, partial η^2^ = 0.26, but not for MP, F(1, 56) = 0.076, *p* = 0.783, partial η^2^ < 0.01. According to Tukey’s HSD test, the MFR group showed significantly higher SO_2_ for t_1_ (M_Diff_ = 10.1, 95%-CI[16.8, 3.39], *p* = 0.043), and for t_2_ (M_Diff_ = 10.3, 95%-CI[16.8, 3.82], *p* = 0.003) compared with the control group.

Descriptive statistics for the group differences after treatment are shown in Table 2 and Figure 4.

### 3.2. Secondary Objectives: Influence of Thoracolumbar Fascia Morphology and Physical Activity on Microcirculation in Lumbar Tissue and Their Correlations

The ICC between the TLFM raters showed excellent agreement, ICC = 0.98, 95%-CI[0.93, 0.98], F(29, 59.8) = 24.4, *p* < 0.0001.

A three-way ANOVA determined no significant interaction between TLFM, group 1: very disorganized (*n* = 5), group 2: somewhat disorganized (*n* = 7), group 3: somewhat organized (*n* = 9), group 4: very organized (*n* = 9), IPAQ, group 1: no sport (*n* = 9), group 2: normal (1–3 h per week; *n* = 8), group 3: sporty (4–7 h per week; *n* = 4), group 4: performer (>8 h per week; *n* = 11), and BMI, on the lumbar myofascial BF, F(3, 21) = 2.13, *p* = 0.117, partial η^2^ = 0.31. There was a significant main effect of TLFM on BF, F(3, 21) = 11.70, *p* < 0.001, partial η^2^ = 0.65, but not of IPAQ, F(2, 21) = 0.72, *p* = 0.551, partial η^2^ = 0.10, or BMI, F(7, 12) = 0.62, *p* = 0.73, partial η^2^ = 0.27. Results of the Tukey’s HSD are shown in Table 3.

A three-way ANOVA determined no significant interaction between TLFM, IPAQ, and BMI on the lumbar myofascial blood oxygen saturation (SO_2_), F(3, 21) = 1.95, *p* = 0.153, partial η^2^ = 0.22. There was a significant main effect of TLFM on SO_2_, F(3, 21) = 4.46, *p* = 0.014, partial η^2^ = 0.39, but not for IPAQ, F(2, 21) = 2.50, *p* = 0.106, partial η^2^ = 0.19, or BMI, F(7, 12) = 1.23, *p* = 0.360, partial η^2^ = 0.42. According to Tukey’s HSD test, the TLFM group 4 showed significantly higher SO_2_ (M_Diff_ = 22.7, 95%-CI[6.0, 39.4], *p* < 0.0049) compared with TLFM group 2.

A three-way ANOVA determined no significant interaction between TLFM, IPAQ, and BMI on the lumbar myofascial rHb, F(3, 21) = 0.23, *p* = 0.875, partial η^2^ = 0.03. There was no significant main effect of TLFM on rHb, F(3, 21) = 0.55, *p* = 0.657, partial η^2^ = 0.07, of IPAQ, F(2, 21) = 0.34, *p* = 0.335, partial η^2^ = 0.03, and of BMI, F(7, 12) = 0.30, *p* = 0.941, partial η^2^ = 0.15. The changes in lumbar microcirculation for the different TLFM groups are shown in Figure 5.

There was a strong, positive correlation between TLFM and IPAQ (d = 0.68, 95%-CI[0.51, 0.85], *p* < 0.0001), and a strong, negative correlation between TLFM and BMI (r(28) = −0.83, *p* < 0.0001).

## 4. Discussion

Examination of the primary objective of the study revealed that MFR treatment had a significant effect on BF (*p* = 0.026). The BF was 31.6% higher in the MFR group than in the PLC group. In the PLC group, the BF decreased actually by −15.4%, which was expected under almost resting conditions with only gentle PLC touch. After the 60 min follow-up measurement, BF increased further by 48.7% in the MFR group compared with the PLC group, whereas BF remained relatively constant in the PLC group. No interaction effects were found for SO_2_ and rHb with respect to MP (all *p* > 0.500), but there were significant group differences between the MFR and PLC groups (all *p* < 0.037). It can be assumed that these changes were mainly caused by the BF increase, which also improves oxygen and erythrocyte supply in myofascial tissue.

MFR treatment induced a much higher force on the treated area through pressure, traction, and stretching of the myofascial tissue than the lighter PLC touch. Therefore, in addition to altering skin receptors, MFR techniques could also stimulate free nerve endings with mechanoceptive, nociceptive, autonomic, and vascular functions in the fascial tissue under the skin. In contrast, the much gentler touch of PLC treatment likely affects only skin receptors [5,16,25,26].

Hoheisel et al. [16] and Mense [5] described mechanosensitive varicosities (axonal widenings storing neuropeptides and neurotrophins) in free nerve endings of TLF innervating their arterioles. Mechanical stimuli release these contents, leading to vasodilation of adjacent arterioles and an increase in BF [5]. The TLF is rhombically pervaded by this dense nerve network [7]. The deformation of the TLF and altered morphology probably also affect blood vessels as well as free nerve endings and could lead to changes in muscle tone, fluid delivery, and other neurological effects that were probably positively influenced by the intervention in this study [25,26,47].

One of the participants from the PLC group suffered from acute nLBP (pain score 10 on the visual analog scale (VAS)) four weeks after the study. In this individual case, the ultrasound image of TLF morphology changed within 10 days after MFR treatment (Figure 6). This was surprising, as it would be expected that a change in tissue structure would take more time [48]. Although this case is not representative, it provides interesting food for thought regarding the mechanisms behind this improvement in morphology. The patient showed some adhesions between fascial layers and an undulating, disorganized structure of the posterior TLF layer, which completely disappeared 10 days after MFR treatment. Strikingly, the patient was also pain-free at this follow-up (pain score 0 on the VAS). The TLF is associated with many muscles (e.g., latissimus dorsi, gluteus maximus, externus oblique, posterior serrati, and erector spinae muscles) and is also stretched and deformed by them [7]. In addition, the TLF is also capable of altering its stiffness by contractile fibroblasts and myofibroblasts within its own tissue [3]. Therefore, adhesions within the fascial layers and imbalanced muscle tensions likely deform both the TLF and the thin, delicate blood vessels and nerve network within it. MFR treatment could counteract some of these mechanisms. Future work investigating the potential ability of the TLF to change its morphology in response to mechanical stimuli is therefore of particular interest.

In this study, the TLFM for the secondary objectives was visually evaluated by three independent raters and classified into four morphology groups according to De Coninck et al. [20]. The ICC between raters was excellent (ICC = 0.98) and consistent with the previous studies on which the criteria for TLFM interpretation were based [20,41].

There were no interactions between TLFM, IPAQ, BMI, and BFD (all *p* > 0.12). Because BMI had no influence on the measurement and the inclusion criteria allowed only a maximum DER and SAT thickness of 7 mm, it can be assumed that BFD originated mainly from the myofascial lumbar tissue and not only from the DER or SAT. Only TLFM had a significant effect on BF and SO_2_ but not on rHb. However, the standard deviation between the more organized groups was large, making the differences questionable in terms of clinical relevance, despite statistical significance. However, even larger mean differences with non-overlapping 95%–CI were observed between disorganized (group 1 or 2) and very organized TLF (group 4) > 74 AU. Whereas BF and SO_2_ increased almost linearly from group 2 to 4, rHb remained relatively constant. This could be interpreted as an increase in blood velocity due to better flow properties of erythrocytes in the vessels [49]. The O2C device measures mainly venous capillary blood. Thus, the results suggest that tissue disorganization in the TLF results in poorer flow properties, especially in the venous drainage system [50].

Some studies have shown that hypoxia following circulation restriction can induce inflammation and apoptosis, leading to fascial degeneration [5,16,22,24]. Hoheisel et al. [16] and Mense [5] found a significantly increased density of substance *p*-positive fibers in inflamed rat TLF. These fibers are generally considered to be nociceptive [51] and could lead to higher pain sensitivity. In addition, there is evidence for a higher density of immunoreactive calcitonin gene-related peptide fibers in inflamed TLFs, which are described as partially mechanoreceptive and partially nociceptive and are most abundant within the TLF [5,16]. This could enhance both nociceptive sensitivity and impaired proprioceptive functions. Therefore, future studies focusing on the relationship between blood circulation, inflammation, and degeneration of the TLF versus altered sympathetic, proprioceptive, and nociceptive innervation are promising in terms of factors contributing to the development of nLBP.

Somers’ D showed a strong positive correlation between TLFM and IPAQ (D = 0.68) and Spearman’s Rho an even stronger negative correlation between TLFM and BMI (r = −0.83). This was expected by the authors, as there were some preliminary results suggesting such an assumption [18,52]. There is also evidence that PA level is associated with nLBP [53]. However, Wakker et al. [54] found no correlations between sex, BMI, IPAQ, and TLF elasticity in 267 healthy participants. This is contrary to the authors’ hypothesis that a more disorganized TLF (which is correlated with BMI and IPAQ) leads to tissue degeneration, lower shear mobility, and also lower elasticity, which has also been shown in previous studies with symptomatic subjects [19,52]. It could therefore be hypothesized that variables such as BMI and IPAQ affect TLF differently in healthy people and people with nLBP. Future work investigating the relationship between variables such as BMI and IPAQ in different modalities (e.g., healthy vs. subjects suffering from nLBP) is strongly recommended.

The study presented here fulfilled the criteria of a randomized control trial. However, due to the fact that the study was conducted within the context of an ongoing manual therapy praxis, the examination and intervention was made by only one investigator. Therefore, the investigator could not be blinded against sham or MFR treatment. However, participants and statisticians were blinded against these modalities. A previous study from the authors, using a similar methodology could therefore reach 8 of 10 points at the PEDro scale reviewed by a meta-analysis [4,55].

Recent research indicates that the IPAQ tends to underestimate or overestimate items related to sitting in its questionnaire [36], while good reliability is attested for other items [56]. Therefore, to avoid an alleged bias of the IPAQ, it is recommended for subsequent work to complement the PA assessment with smart trackers that provide information on sedentary activities [38].

Today, no general expert consensus on TLFM exists. The assessment of this variable in this investigation is based on two studies with relatively small cohort sizes of 30 investigated study participants and 30 observers in one [20] and 30 participants and one observer in the other [43]. However, both studies found excellent ICC between and within assessors. In this work, TFLM was studied by three blinded, experienced raters. They also agreed with an excellent ICC of 0.98. It can be assumed that at least very disorganized and very organized TFL can be distinguished from each other, which could be confirmed by the significant results between these groups.

The O2C device measures at a tissue depth of up to 8 mm, depending on the fiber optic probe used. It cannot be generally assumed that this arrangement allows BFD to be derived exclusively from the TLF. Participants with a DER and SAT greater than 7 mm thick were excluded. To further minimize the influence of increased adipose tissue, a three-way ANOVA was performed to additionally determine the influence of BMI. Because there were no such interactions between BFD and BMI, and based on the dense optical properties of TLF, it is likely that most BFD originated from fascial tissue rather than muscle tissue. However, it is also possible that some of the BFD originates from muscle tissue, which must be added to the TLF BFD. Therefore, in this study, the term “myofascial” BFD was used throughout and no differentiation was made between TLF and muscle.

The results of this study should be viewed in light of the objective as, to the authors’ knowledge, this is the first study to examine the effects of TFLM, PA, pain, and MFR on BFD and to compare them with a sham intervention. The results show that TFLM, IPAQ, and BMI have a critical impact on TLF BFD. The authors further hypothesize that a possible mechanism behind the treatment effects may be that higher pressure, traction, and stretching stimulates the free nerve endings with mechanoceptive, nociceptive, autonomic, and vascular functions in the fascial tissue under the skin, as opposed to the much gentler touch of sham treatment.

## 5. Conclusions

MFR treatment significantly increased BF by 31.6% immediately after the procedure and by 48.7% at 60 min follow-up compared with the PLC group. SO_2_ and rHb also increased in the MFR group following BF. TLFM, IPAQ, and BMI showed strong correlations with BFD. Circulatory restrictions could lead to hypoxia-induced inflammation, which likely causes pain and impaired proprioceptive function and may contribute to the development of nLBP.

Dysfunction of blood vessels and free nerve endings, which are presumed to be positively associated with TFLM, were likely positively affected by the intervention in this study. This may offer a new perspective for the prevention and also the treatment of low back pain by broadening the focus to the fascial organization, but it needs to be further clarified in subsequent studies.

## Figures and Tables

**Figure 1 jcm-12-01248-f001:**
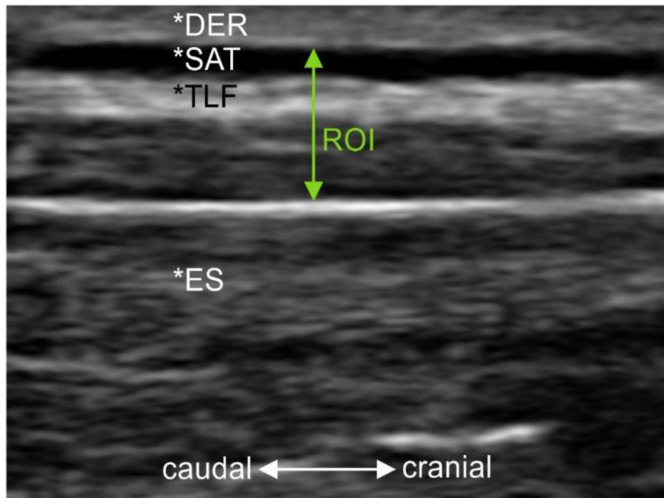
Anatomical orientation and delineation of the zones rated. *DER, dermis; *SAT, subcutaneous adipose tissue; *TFL, thoracolumbar fascia; *ES, erector spinae muscle; ROI, region of interest, zones rated.

**Figure 2 jcm-12-01248-f002:**
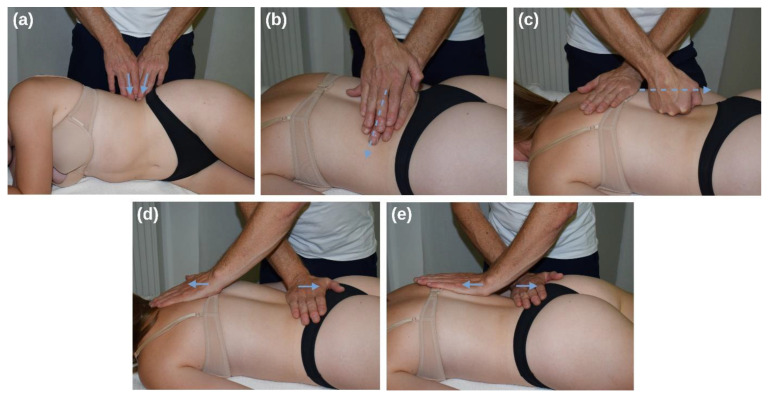
Myofascial release and placebo treatment at the TLF. (**a**) Sustained manual pressure to the lateral. (**b**) Lateral stretching of the TLF. (**c**) Longitudinal glide along the lumbar paravertebral muscles. (**d**) Longitudinal stretch of the TLF. (**e**) Unilateral longitudinal stretch of the TLF. Blue arrows show the direction of tissue stretching in the myofascial release treatment. In the placebo treatment, the hands were instead left in place with minimal pressure.

**Figure 3 jcm-12-01248-f003:**
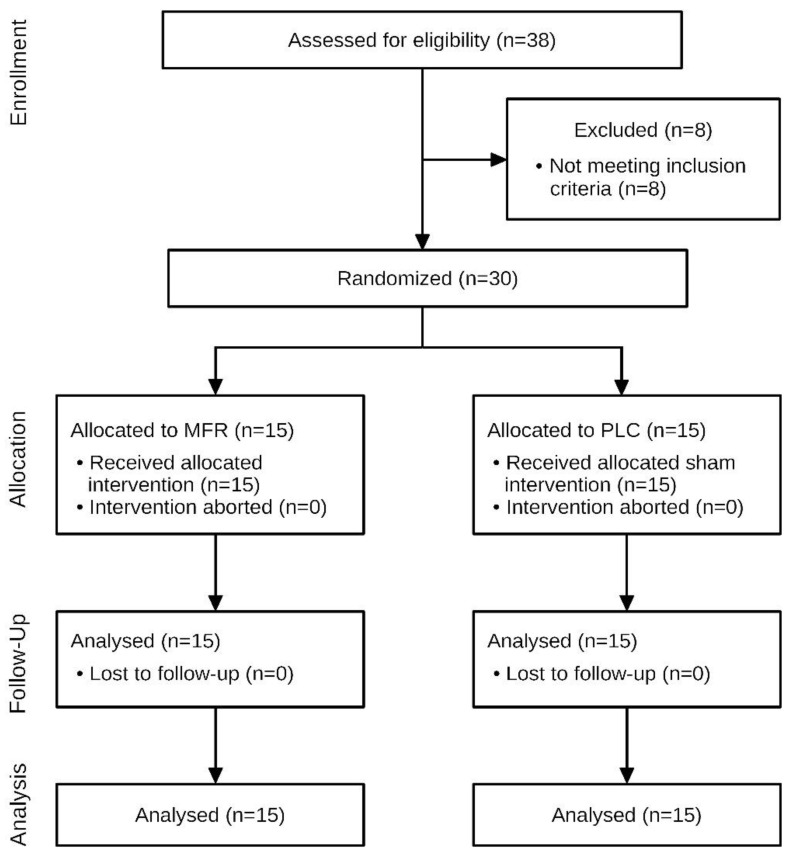
Flow diagram of the study. *n*, number; MFR, myofascial release; PLC, placebo.

**Figure 4 jcm-12-01248-f004:**
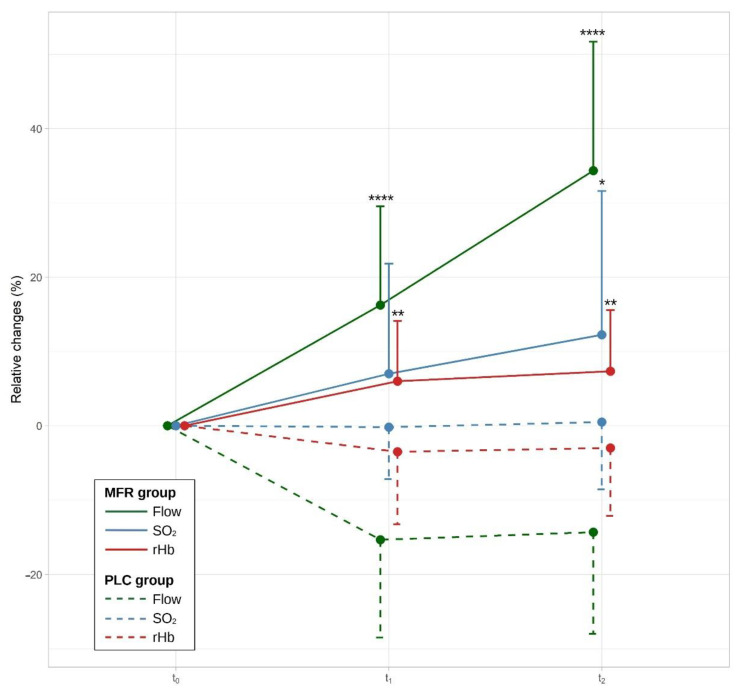
Relative changes in percent compared to baseline measurement. For better readability, the error bars are only shown on one side and represent the standard deviation. t_0_, baseline measurement; t_1_, measurement after treatment; t_2_, measurement 40 min after treatment; SO_2_, oxygen saturation; rHb, relative hemoglobin. Group differences, significant at the level * *p* < 0.05, ** *p* < 0.01, **** *p* < 0.0001.

**Figure 5 jcm-12-01248-f005:**
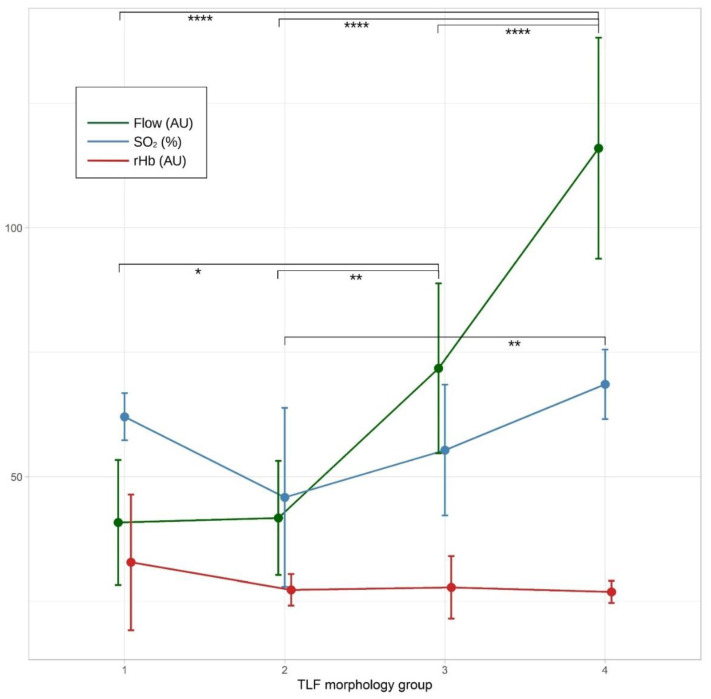
Changes in lumbar microcirculation between the thoracolumbar fascia morphology groups (group 1: very disorganized (*n* = 5), group 2: somewhat disorganized (*n* = 7), group 3 somewhat organized (*n* = 9), group 4: very organized (*n* = 9)). SO_2_, oxygen saturation; rHb, relative hemoglobin; AU, arbitrary units. Error bars represent the standard deviation. Significant at the level * *p* < 0.05, ** *p* < 0.01, **** *p* < 0.0001.

**Figure 6 jcm-12-01248-f006:**
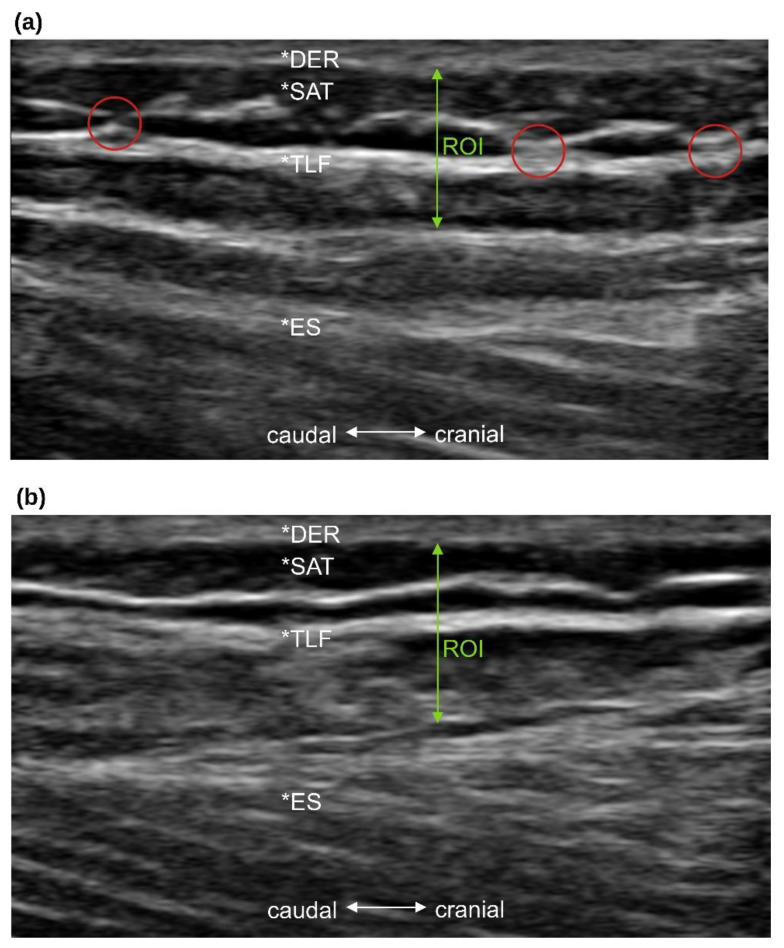
Thoracolumbar fascia of a patient with acute lumbar back pain. The red circles show adhesions 24 h after the lumbago, likely causing the posterior layer to take on an undulating shape that was classified as “somewhat disorganized” (**a**). 10 days after myofascial release treatment, the adhesions disappeared and the pain subsided completely. The fascia was then rated as “very organized” (**b**). *DER, dermis; *SAT, subcutaneous adipose tissue; *TFL, thoracolumbar fascia; *ES, erector spinae; ROI, region of interest, zones rated.

**Table 1 jcm-12-01248-t001:** Baseline characteristics.

Baseline Characteristics	Participants (*n* = 30) Mean ± SD
Gender (men/women)	15/15
Age (years)	40.5 ± 14.1
Height (m)	1.73 ± 0.1
Weight (kg)	69.0 ± 11.5
BMI (kg/m^2^)	23.0 ± 3.2
Physical activity (IPAQ)	
High, *n* (%)	11 (36)
Moderate, *n* (%)	12 (40)
Low, *n* (%)	7 (23)

SD, standard deviation; *n*, number.

**Table 2 jcm-12-01248-t002:** Descriptive statistics.

					95% Confidence Interval of Mean	
Outcome	Time	Group	*n*	Mean ± SD	Lower Bound	Upper Bound
Flow	t1	MFR	15	16.20 ± 13.30	8.84	23.56
		PLC	15	−15.40 ± 16.10	−22.68	−8.12
	t2	MFR	15	34.40 ± 17.40	24.79	44.01
		PLC	15	−14.30 ± 13.70	−21.87	−6.73
SO_2_	t1	MFR	15	6.78 ± 14.80	−1.42	14.98
		PLC	15	−0.19 ± 7.00	−4.06	3.68
	t2	MFR	15	12.20 ± 19.40	1.50	22.90
		PLC	15	0.51 ± 9.08	−4.52	5.54
rHb	t1	MFR	15	6.56 ± 8.11	2.10	11.08
		PLC	15	−3.51 ± 9.74	−8.91	1.89
	t2	MFR	15	7.33 ± 8.24	2.76	11.90
		PLC	15	−3.00 ± 9.14	−8.06	2.06

Values show relative changes in percent compared to baseline measurement. SO_2_, oxygen saturation; rHb, relative hemoglobin; AU, arbitrary units; SD, standard deviation; *n*, number.

**Table 3 jcm-12-01248-t003:** Influence of thoracolumbar fascia morphology on blood flow in lumbar tissue.

			95% Confidence Interval of Mean		
TLF Groups		Mean (AU)	Lower Bound	Upper Bound	*p* (Adj.)
1 (*n* = 5)	2 (*n* = 7)	0.91	−26.7	28.5	1
1 (*n* = 5)	3 (*n* = 9)	31.0	4.66	57.3	0.0167
1 (*n* = 5)	4 (*n* = 9)	75.2	48.9	102	<0.0001
2 (*n* = 7)	3 (*n* = 9)	30.1	6.28	53.8	0.0093
2 (*n* = 7)	4 (*n* = 9)	74.3	50.5	98.1	<0.0001
3 (*n* = 9)	4 (*n* = 9)	44.2	22.0	66.5	<0.0001

Tukey’s HSD is shown. TLF thoracolumbar fascia morphology, AU arbitrary units, SD standard deviation, *n* number, adj. adjusted.

## Data Availability

Data can be made available by the authors upon request.

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
