# Peer review of "Immediate Effects of Myofascial Release Treatment on Lumbar Microcirculation: A Randomized, Placebo-Controlled Trial"

_jcm, 2023, doi:10.3390/jcm12041248_

Round 1

Reviewer 1 Report

Hello dear editor.

I am pleased to receive an interesting article entitled "Influence of sports activity, thoracolumbar fascia morphology and myofascial release treatment on lumbar microcirculation: a randomized, placebo-controlled trial" for refereeing. It is clear that the authors have tried hard to meet the standards of scientific writing.  The following items are suggested to improve its quality:

1- Please clearly mention in the title and summary of the article that the immediate effect of a therapeutic intervention session has been measured.

2-Please mention the blood flow measurement method (Doppler ultrasound) in the summary of the article as well as in the method section (page 4).

3- Please write in the introduction section that the follow-up was done with what scientific background?

4- In the characteristics of the samples entering the research, as well as in the summary section of the article, it is mentioned that the people did not have pain during the research. So please explain the finding of a strong negative correlation between pain (r=-0.648) and blood flow.

5- On page 2, the last paragraph of the sentence "Therefore, Wu et al. [28] showed improvement in pain and physical function." seems to be repeated according to the meaning of the previous and subsequent contents. Please delete or rewrite this sentence.

6- Usually, the amount of experience of researchers in measuring a research parameter is not mentioned in many articles. In the text of your article, recheck the sentence "The subjects received the respective group-specific intervention from an osteopath who had more than 15 years of professional experience in manual therapy and a master’s degree. " on page 4 of the first paragraph and the sentence "Three different raters with at least 10 years of experience in evaluating ultrasound images (two physicians and one osteopath with a master's degree) independently assessed TLFM in a blinded fashion with respect to the study participants and to the other raters." on page 5 of the last paragraph. Is mentioning these sentences consistent with the rules of the JCM?

7- Based on table 1 on page 7 and title 3.1, it is suggested to change the words "Sports activity" in the title and text of the article to "Physical activity".

8- Please compare heading 3.2 on page 8 as “3.2. Differences between Myofascial Release treatment and placebo intervention” with heading 3.3 on page 9 as “3.3. Differences between Myofascial Release treatment and placebo intervention”. Are these two titles and the content of a paragraph below them different?

9- Please state in the findings section how many samples you had in each of the 4 categories of thoracolumbar fascia organization?

10- Related to Figure 4 on page 10, please in the discussion section justify the Error bar largeness of the flow (AU) parameter especially in groups 3 and 4.

11- It is recommended to mention all the findings in the relevant section and move figures 4-6 as well as table 3 from the discussion section to the findings section.

Author Response

Dear Reviewer,

We were very impressed with the thorough review and the many valuable recommendations and improvements we received from you. We are convinced that this will significantly improve the quality of our article. We have revised our article in this regard, following your recommendations step by step. Below we comment point by point on your comments.

  1. Please clearly mention in the title and summary of the article that the immediate effect of a therapeutic intervention session has been measured.
    We changed the title and clearly mentioned "immediately". To emphasize this we have separated into primary and secondary objectives and also highlighted that we measured immediate impact in the summary. Lines 2, 21-26, 30-35, order of primary and secondary objectives in the manuscript.
  2. Please mention the blood flow measurement method (Doppler ultrasound) in the summary of the article as well as in the method section (page 4).
    We added the information. Line 30, 260, 261.
  3. Please write in the introduction section that the follow-up was done with what scientific background?
    We added a reference and described our intent. Lines 120-123.
  4. 4- In the characteristics of the samples entering the research, as well as in the summary section of the article, it is mentioned that the people did not have pain during the research. So please explain the finding of a strong negative correlation between pain (r=-0.648) and blood flow.
    That was a mix up. It shouldn't be "pain" but "physical activity". We corrected that. Line 35.
  5. On page 2, the last paragraph of the sentence "Therefore, Wu et al. [28] showed improvement in pain and physical function." seems to be repeated according to the meaning of the previous and subsequent contents. Please delete or rewrite this sentence.
    We deleted the sentence. Line 104.
  6. Usually, the amount of experience of researchers in measuring a research parameter is not mentioned in many articles. In the text of your article, recheck the sentence "The subjects received the respective group-specific intervention from an osteopath who had more than 15 years of professional experience in manual therapy and a master’s degree. " on page 4 of the first paragraph and the sentence "Three different raters with at least 10 years of experience in evaluating ultrasound images (two physicians and one osteopath with a master's degree) independently assessed TLFM in a blinded fashion with respect to the study participants and to the other raters." on page 5 of the last paragraph. Is mentioning these sentences consistent with the rules of the JCM?
      Indeed, we can agree with your argument that mentioning the researcher's experience might bias the readers, but in our opinion, experiences of the therapist in an interventional study should be reported. Also, the experience of a sonographer to provide the readers with information about a clinical background. Studies have also shown that examiner experience is a crucial factor in scientific assessments[1]. We also searched for intervention studies published in JCM and found that most of them reported on the therapist's experience.
  7. Based on table 1 on page 7 and title 3.1, it is suggested to change the words "Sports activity" in the title and text of the article to "Physical activity".
    We followed your recommendation and changed the expression.
  8. Please compare heading 3.2 on page 8 as “3.2. Differences between Myofascial Release treatment and placebo intervention” with heading 3.3 on page 9 as “3.3. Differences between Myofascial Release treatment and placebo intervention”. Are these two titles and the content of a paragraph below them different?
    That was a formatting mistake. We corrected that.
  9. Please state in the findings section how many samples you had in each of the 4 categories of thoracolumbar fascia organization?
    We have already provided this information. However, it is now reorganized into Section 3.2, Table 3 and Figure 5. Lines 353-361, 401, 431-435.
  10. Related to Figure 4 on page 10, please in the discussion section justify the Error bar largeness of the flow (AU) parameter especially in groups 3 and 4.
    We added the information what the error bars in the figure represent (standard deviation) and discussed the possible meaning in the discussion section. Lines 434, 548-553.
  11. It is recommended to mention all the findings in the relevant section and move figures 4-6 as well as table 3 from the discussion section to the findings section.
       We restructured the paper according to our primary and secondary objectives and tried to arrange tables and figures in a format-appropriate order.

[1] van Trijffel, E.; Anderegg, Q.; Bossuyt, P.M.M.; Lucas, C. Inter-Examiner Reliability of Passive Assessment of Intervertebral Motion in the Cervical and Lumbar Spine: A Systematic Review. Manual Therapy 2005, 10, 256–269, doi:10.1016/j.math.2005.04.008.

Reviewer 2 Report

Specific Recommendations

- Obviously, the importance of the article could not be emphasized enough in the analysis and result section. When the abstract section is examined, it is thought that the study is a cross-sectional study. For this reason, In this study, the results of 2-way analysis of variance should be emphasized instead of correlation analyses. Abstract, results and discussion sections should be organized in the relevant fields.

Author Response

Dear Reviewer,

Thank you for the review and the valuable recommendations we received from you. We are convinced that this will improve the quality of our article. We have revised our article with this in mind, following your recommendations. In the following, we will address your comments point by point.

  1. Obviously, the importance of the article could not be emphasized enough in the analysis and result section. When the abstract section is examined, it is thought that the study is a cross-sectional study. For this reason, In this study, the results of 2-way analysis of variance should be emphasized instead of correlation analyses.
    This is a valuable comment. We have reorganized our paper and revised the objectives into primary and secondary. The primary objective was to compare the groups using an RCT design. Secondarily, we compared TLFD, IPAQ, and BMI to blood flow using an ANOVA. TLFD and IPAQ are ordinally scaled (at least one interval-scaled dependent variable is required for an ANOVA), so Somers’ d was used to find putative correlations.
  2. Abstract, results and discussion sections should be organized in the relevant fields.
        We restructured the paper according to our primary and secondary objectives and organized the sections in the relevant areas.

Round 2

Reviewer 2 Report

The authors have made request adjustments. Congratulations.